# Reduced Striatal Dopamine Transporter Availability and Heightened Response to Natural and Pharmacological Stimulation in CCK-1R-Deficient Obese Rats

**DOI:** 10.3390/ijms24119773

**Published:** 2023-06-05

**Authors:** Sevag Hamamah, Andras Hajnal, Mihai Covasa

**Affiliations:** 1Department of Basic Medical Sciences, College of Osteopathic Medicine, Western University of Health Sciences, Pomona, CA 91766, USA; sevag.hamamah@westernu.edu; 2Department of Neural and Behavioral Sciences, College of Medicine, The Pennsylvania State University, Hershey, PA 17033, USA; ahajnal@psu.edu; 3Department of Biomedical Sciences, College of Medicine and Biological Science, University of Suceava, 720229 Suceava, Romania

**Keywords:** dopamine, OLETF, obesity, sucrose, amphetamine, locomotor activity

## Abstract

Alterations in dopamine neurotransmission are associated with obesity and food preferences. Otsuka Long-Evans Tokushima Fatty (OLETF) rats that lack functional cholecystokinin receptor type-1 (CCK-1R), due to a natural mutation, exhibit impaired satiation, are hyperphagic, and become obese. In addition, compared to lean control Long-Evans Tokushima (LETO) rats, OLETF rats have pronounced avidity for over-consuming palatable sweet solutions, have greater dopamine release to psychostimulants, reduced dopamine 2 receptor (D2R) binding, and exhibit increased sensitivity to sucrose reward. This supports altered dopamine function in this strain and its general preference for palatable solutions such as sucrose. In this study, we examined the relationship between OLETF’s hyperphagic behavior and striatal dopamine signaling by investigating basal and amphetamine stimulated motor activity in prediabetic OLETF rats before and after access to sucrose solution (0.3 M) compared to non-mutant control LETO rats, as well as availability of dopamine transporter (DAT) using autoradiography. In the sucrose tests, one group of OLETF rats received ad libitum access to sucrose while the other group received an amount of sucrose equal to that consumed by the LETO. OLETFs with ad libitum access consumed significantly more sucrose than LETOs. Sucrose exerted a biphasic effect on basal activity in both strains, i.e., reduced activity for 1 week followed by increased activity in weeks 2 and 3. Basal locomotor activity was reduced (−17%) in OLETFs prior to sucrose, compared to LETOs. Withdrawal of sucrose resulted in increased locomotor activity in both strains. The magnitude of this effect was greater in OLETFs and the activity was increased in restricted compared to ad-libitum-access OLETFs. Sucrose access augmented AMPH-responses in both strains with a greater sensitization to AMPH during week 1, an effect that was a function of the amount of sucrose consumed. One week of sucrose withdrawal sensitized AMPH-induced ambulatory activity in both strains. In OLETF with restricted access to sucrose, withdrawal resulted in no further sensitization to AMPH. DAT availability in the nucleus accumbens shell was significantly reduced in OLETF compared with aged-matched LETO. Together, these findings show that OLETF rats have reduced basal DA transmission and a heightened response to natural and pharmacological stimulation.

## 1. Introduction

Obesity is one of the most pervasive chronic diseases with a high prevalence both in the United States and worldwide, and is projected to affect over one billion individuals by 2030 [1]. Its etiology is multifactorial and complex, leading to an imbalance between homeostatic and non-homeostatic mechanisms that are ordinarily in place to maintain energy balance and body weight [2]. As such, increased palatability to the readily available foods rich in sugar and fats has become a driving factor of motivational eating even in a repleted or excess state of energy storage, overriding metabolic and homeostatic mechanisms. This excess caloric intake is regulated by central mechanisms of reward circuitry, of which the mesocorticolimbic dopaminergic pathway plays an important role [3]. Current studies support the hypothesis that obese subjects exhibit deficits in the reward circuitries and a hyposensitivity of dopaminergic system [4,5]. For example, obese individuals have a more reduced dopamine (DA) D2 receptor (D2R) binding than normal weight subjects and display reduced striatal activation when consuming palatable foods [6,7]. Reduced striatal dopamine transmission in obese individuals mirrors findings in persons with substance use and addiction [8,9,10] underlying the reward-deficiency hypothesis [11,12]. This postulates that blunted DA signaling in obese subjects leads to increase reward-seeking to compensate for the DA deficits, a behavior that may also result in chronic over-eating particularly stimulated by highly palatable “junk foods” overtime, resulting in obesity and associated metabolic disorders [13].

Among obese rodent models used to characterize dopaminergic malfunctioning potentially driving excessive food intake, the Otsuka Long-Evans Tokushima Fatty (OLETF) rats have been used to assess mesolimbic DA transmission in absence of the cholecystokinin (CCK) type 1 receptors (CCK-1R) [14]. CCK-1Rs are present in both the gut and the brain, with CCK-producing neurons in the brain being widely distributed throughout the mesolimbic system, as well as other brain structures including the substantia nigra, area postrema, medial pre-optic area, arcuate nucleus, and cortex [15]. CCK-1R expression in these regions has been shown to be intricately correlated with the dopaminergic system, with CCK/DA being found to not only be co-localized, but also co-released [16,17]. Indeed, accumulated data from our labs [7,13,18,19,20,21] and others have shown that OLETF rats have a dysfunctional DA signaling that may be contributory to hyperphagia, and, in turn, their obesity. In addition, OLETF rats exhibit greater preference for sweet and fatty solutions compared to the non-mutant control Long-Evans Tokushima Otsuka (LETO) rats and display increased sensitivity to sucrose reward [19]. Specifically, OLETF rats perform more licks in brief-intake tests to various sweet-tasting carbohydrates and non-carbohydrate solutions, and expend more effort for seeking and obtaining sucrose reinforcement in a progressive ratio operant task. Such behaviors are consistent with changes in the mesolimbic DA functions in OLETF rats as evidenced by the following findings: OLETF rats have an increased basal DA levels [22]; are more sensitive to peripheral administration of D1R or D2R antagonists in reducing sucrose intake [13]; have a significantly lower D2R binding in the nucleus accumbens (NAcc) [7]; have enhanced amphetamine-induced increases in NAcc DA release and are less sensitive to the pre-pulse inhibition effects of sucrose and amphetamine relative to the LETO controls [23]. Together, these studies indicate a generalized dopaminergic hyposensitivity in OLETF rats characteristic of deficits in the brain reward circuitry leading to their strong avidity for palatable foods. However, how access to palatable foods such as sweets and how their withdrawal effects might influence reward sensitivity in the OLETF rats has not been studied.

Therefore, to examine the severity of the dopaminergic dysfunction in OLETF rats exposed to chronic sweet stimulation, we measured basal and amphetamine (AMPH)-induced locomotor activity in prediabetic OLETFs compared to age-matched LETOs. These values were measured prior to and after a 1- and 3-week ad libitum or restricted (clamped to the amount consumed by LETO controls) access to palatable sucrose solution. D-amphetamine has been used as a pharmacological tool to assess sensitivity of the dopaminergic system due to its dopamine-releasing properties, and consequent elevations of extra synaptic dopamine levels [24,25]. To examine whether changes in dopamine transporter (DAT) contributes to heighten DA sensitivity in OLETF, we also used autoradiography (ARG) to quantify DAT availability in the NAcc and additional nigrostriatal brain areas. Finally, we assessed glucose and plasma insulin, leptin, and corticosterone levels known for their central effects on food intake and DA neurotransmission.

## 2. Results

### 2.1. Effects of Sucrose and Withdrawal on Basal Locomotor Activity

Prior to sucrose access, basal locomotor activity is mildly reduced in OLETFs (−17%, *p* < 0.05) compared to LETOs (Figure 1A). Sucrose access exerts a biphasic effect on basal activity in both strains, i.e., reduced activity for ~1 week followed by normal or increased activity throughout weeks 2 and 3 (Figure 1B). The magnitude of this effect, however, is a function of the amount of sucrose consumed, with a reduction in activity during week 1 being the most pronounced in OLETFs with ad lib sucrose access and not statistically significant in the LETO group as well as OLETFs with restricted sucrose access. Withdrawal of sucrose results in increased activity in both strains, however, the magnitude of this effect, again, is greater in OLETF rats, and differential with respect to sucrose access. In contrast to sucrose’s effect on first week activity, withdrawal increases motor activity more in restricted-access than in ad-libitum-access OLETFs (Figure 1B).

### 2.2. Effects of Sucrose and Withdrawal on AMPH-Induced Locomotor Activity

Under basal conditions (i.e., no sucrose), a low dose of AMPH results in slightly increased ambulatory locomotor activity (+16%, *p* < 0.05) in OLETF compared to LETO rats.

Sucrose access augments AMPH-responses in both strains. Similarly, 1 week withdrawal of sucrose sensitizes AMPH-induced ambulatory activity in both strains. However, in OLETFs compared to LETOs, sucrose access results in a greater sensitization to AMPH during week 1, an effect that is a function of the amount of sucrose consumed (i.e., ad lib OLETF > clamped OLETF). Sucrose withdrawal results in no further sensitization to AMPH (i.e., above-baseline response relative to LETO) in restricted-sucrose-access OLETFs (Figure 2).

### 2.3. Dopamine Transporter Binding Assays

The results from the ^125^I-RTI-55 quantitative autoradiography are depicted in Figure 3. They reveal a 20% reduction (*p* < 0.05) in DAT binding in the shell but not the core region of the NAcc of 14 weeks old naïve, prediabetic OLETFs compared with age-matched LETOs. This effect is specific to the caudal medial shell of NAcc of the OLETF compared to age-matched lean LETO rats. None of the other brain areas investigated show statistically significant difference in binding between the strains (Figure 3A). Results also show that DAT-binding negatively correlates with body mass (R = −0.7491, *p* < 0.001) (Figure 3B).

### 2.4. Sucrose, Food Intake, and Body Weight

OLETF rats with ad libitum access to sucrose consume significantly more sucrose than LETO rats (132.8 ± 14.3 vs. 85 ± 8.25 mL/24 h) and gain more weight (55.4 ± 6.6 vs. 34 ± 4.2 g) during the 3 week sucrose access period (Figure 4A,C, respectively). In addition, unlike sucrose-fed LETO rats, neither ad libitum nor restricted sucrose-fed OLETF rats compensate for calories by reducing their concurrent intake on chow (Figure 4B). When sucrose is removed (boxed area in Figure 4C), both OLETF groups lose less weight than LETO groups.

### 2.5. Blood Glucose, Plasma Corticosterone, Leptin and Insulin Levels

Relative to age-matched LETOs, OLETFs reveal significantly higher blood glucose, plasma insulin, and leptin levels, but no difference in plasma corticosterone between strains (Figure 5).

## 3. Discussion

### 3.1. Effects of Sucrose and Withdrawal on Basal Locomotor Activity

Under basal conditions, i.e., prior to sucrose administration, compared to age-matched LETO, the OLETF rats show an overall reduced locomotor activity (Figure 1A). These findings are in agreement with data from other laboratories [26,27,28], and may be explained by the absence of CCK-1R regulation of firing rate and dopamine release in the VTA and NAcc, respectively [22,29]. Further, CCK is co-released with DA within other brain structures associated with motor activity such as the substantia nigra and the dorsal striatum [30], contributing to DA bioavailability in the nigrostriatal pathway and, therefore, motor activity. It is important to note that, in the absence of CCK-1R activity, the balance may also be shifted to the CCK-2R function. Stimulation of CCK-2R decreases dopamine release in the NAcc while CCK-1R generally promotes increased dopamine release [31]. Though no studies directly show these effects in the OLETF strain, antagonist and agonist studies elucidate these contrasting effects on DA activity. For example, CCK-2R knockout mice exhibit hyper locomotor activity while CCK2 agonists show impairment in similarly assessed behaviors, indicating that CCK2R stimulation has negative modulatory implications on DA neurotransmission [32]. However, both receptor types are shown to have varying affinity and heterogeneity, which further complicates our understanding of these processes [33,34], though the overall hypodopaminergic effect on locomotor activity is shown to be consistent with prior data in this field, which may be partially due to unopposed CCK-2R-mediated DA antagonist-like activity.

### 3.2. Effect of Sucrose and Withdrawal on AMPH-Induced Locomotor Activity

A low dose administration of AMPH results in slightly increased ambulatory activity in OLETF compared to LETO rats at baseline. These findings are somewhat different from those of Feifel et al., who found no significant changes in locomotor activity in the two strains [35]. However, we used an open-field arena instrument compared to the locomotor cages utilized by Feifel et al. Nevertheless, DA concentration is increased after AMPH administration in the NAcc and the caudate/putamen of OLETF rats compared to LETO [23]. This is in line with increased CCK and overflow of DA in the NAcc following AMPH in wild-type murine models, demonstrating the importance of CCK-1R in dopaminergic transmission [36,37]. CCK receptors mediate pre-pulse inhibition of DA release in the neostriatum and mesolimbic pathway [38,39], an effect attenuated in OLETF mice [35]. The degree of pre-pulse inhibition has been shown to be a predictor of CCK-mediated effects on mesolimbic DA and AMPH-induced hyperlocomotion, with disruption of pre-pulse inhibition being primarily mediated by the CCK-1R following stimulant administration [40]. Therefore, the absence of CCK-1R in the OLETF strain is likely contributing to our observed differences in ambulatory activity in response to AMPH administration in the two strains.

Sucrose access exerts a biphasic effect on basal activity in both OLETF and LETO strains, with reduced activity for approximately one week, which progresses to normal or increased activity throughout weeks two and three. The magnitude of this effect is a function of the amount of sucrose consumed, i.e., reduction in activity during the first week is the most pronounced and only statistically significant in OLETFs with ad libitum sucrose access. Similarly, in OLETFs compared to LETOs, sucrose access results in a greater sensitization to AMPH during the first week, an effect that is also found to be a function of the amount sucrose consumed (i.e., ad lib OLETF > restricted access OLETF). Prior to sucrose sensitization, OLETF rats exhibit a decreased ambulatory response to AMPH. Therefore, our study shows that access to sucrose prior to AMPH stimulation plays an important role in sensitization of OLETF rats, potentially due to its effects on the dopaminergic system and lack of CCK-1R. Access to palatable foods such as sugars has been shown to sensitize D1R and opioid mu-1 receptor binding in the NAcc, an effect similar to drugs of abuse [41,42]. Furthermore, PET imaging studies show that both palatable foods and AMPH consistently blunt D2R signaling in the striatum [7,43,44]. Of special importance to the current study, CCK-receptor-deficient mice also show decreased striatal D2R expression after sugar sensitization [45]. Therefore, providing rats with freely available sucrose, repeated stimulation of Nacc DA receptors, results in increased sensitization to other, pharmacological, DA-stimulating agents such as AMPH. Taken together, the effect of sucrose administration on DA transmission along with the absence of functional CCK receptors may explain the findings of increased cross-sensitization in OLETF rats to AMPH as shown in our study. It should be pointed out that transmission is not only a function of uptake, which is the immediate focus of our study, but also is modulated by release and receptor sensitivity, as supported by the above-mentioned research along with the new findings presented here.

Withdrawal of sucrose results in increased activity to a greater magnitude in OLETFs in response to AMPH. However, in OLETFs compared to LETOs, sucrose access results in a greater sensitization to AMPH during week 1, an effect that is also a function of the amount of sucrose consumed ad libitum being greater than restricted access. Sucrose withdrawal results in no further sensitization to AMPH (i.e., above-baseline response relative to LETO) in restricted-sucrose-access OLETFs. It is known that palatable sweets such as sucrose produce reinforcing and reward-mediated behaviors that stem from physical dependence, therefore, removing access should precipitate withdrawal symptoms. However, the extent of withdrawal and observed differences between the two strains and between ad libitum and restricted access rats may be at least partially explained by pre-pulse inhibition and its relation to the CCK-1R [35,40,46,47]. Prior data from our laboratory also show that sucrose access in CCK-1-deficient rats decreases sensitivity to the pre-pulse inhibition effects seen in their non-mutant counterparts [46]. Therefore, it would make sense that AMPH stimulation following heightened sensitization would increase relative locomotor activity in OLETF rats, particularly after withdrawal, as shown in our study.

### 3.3. DAT Binding in the NAcc Shell

Here, we show a 20% reduction in DAT availability in the striatum of 14 weeks old naïve, obese OLETFs compared with age-matched LETOs at baseline, specifically in the caudal medial shell of NAcc (Figure 3A). DAT is a critical modulator of dopaminergic transmission that helps with recycling released DA from the extracellular compartment to the presynaptic terminal to ultimately be metabolized or re-packaged into releasable vesicles [48]. Whereas the difference in DAT expression between OLETF and LETO rats limited to the NAcc shell is somewhat surprising, recent findings also support regionally specific differences in DAT binding and availability that may be explained by the functional interplay between CCK-1R and DA. For example, administration of a CCK analogue favors expression of DA and its metabolites, DOPAC and HVA, in the NAcc shell over the core [49]. Additionally, in the same study, treatment with devazepide, a CCK-1 antagonist, reverses these effects, potentially due to unopposed CCK-2 receptor activity. The CCK-2Rs have been found to be significantly more abundant in the NAcc core as compared to the shell while the opposite is true for CCK-1R [50,51]. Studies evaluating the infusion of CCK into the NAcc shell show increased DA release in the area, due to DA agonist-like behaviors of CCK-1R stimulation [52]. As such, decreased DA bioavailability and release in the NAcc secondary to a lack of CCK-1R-mediated DA agonist-like activity in OLETF rats would correlate with decreased DAT expression in this region, as there would be less DA available. Further, DAT in the NAcc shell is more sensitive to psychomotor stimulation, as compared with the core [53]. This would support our findings of differential responses to AMPH stimulation in the two strains as well. Since we used a normal diet, our findings suggest that the obesogenic diet may not be the main factor in causing diminished DAT availability. An earlier study showed reduced DAT in the dietary obese model that affected all mesolimbic dopaminergic regions of the brain, compared with our study where the effect was localized to the NAcc shell [54]. It is important to note, however, that overall expression of DAT in these regions is not necessarily reduced, but rather obesity may interfere with DAT trafficking to the synaptic membranes, reducing the rate of dopamine reuptake [54]. Taken together, our findings contribute to the strong evidence showing that multiple aspects of the dopaminergic system, including transporters and receptors, are altered in hyperphagic, obese OLETF rats, all of which can potentially contribute to increased preference for highly palatable and high calorie foods due to altered compensatory regulation.

### 3.4. Sucrose, Food Intake and Body Weight

OLETF rats with ad libitum access to sucrose display a significant increase in sucrose intake and weight gain when compared to their LETO counterparts (Figure 4). This is in extension of our previous work showing that OLETF rats express increased oral sensitivity to sweet tastes accompanied by a decreased intestinal sensitivity to inhibitory feedback from digested nutrients [20]. This can be partially explained by the effects of sucrose and other highly palatable foods, leading to alterations in the mesolimbic DA receptor expression and sustained increased extracellular DA concentrations in the NAcc [55,56,57]. Further, gut CCK-1Rs mediate inhibition of gastric emptying via vagal afferents [58], which serves as an important negative feedback mechanism in response to hyperphagia [59]. However, there is a strong orosensory contribution to hyperphagia and weight gain in addition to postprandial negative feedback mechanisms in the OLETF strain [20]. Additionally, our findings show that neither ad-lib-fed nor restricted-sucrose-fed OLETF rats compensate by reducing their concurrent intake of chow (Figure 4). Interestingly, CCK-2R activity may play a role in mediating central dopaminergic system in regulating food intake. Indeed CCK-2R antagonists reduce food intake, indicating that shifts toward CCK-2R activity in OLETF rats may lead to hyperphagia via downstream DA signaling pathways [60].

### 3.5. Blood Glucose, Plasma Corticosterone, Leptin and Insulin Levels

OLETFs have elevated blood glucose, plasma insulin, and leptin levels, an effect consistent with the OLETF phenotype and in agreement with other findings [61,62], though the exact mechanisms behind hyperinsulinemia are not fully clear. It has been suggested that decreased adiponectin in OLETF rats may contribute to hyperinsulinemia [63,64]. Of particular relevance to our discussion, insulin resistance in brain neurons and glia has been shown to be associated with dopaminergic dysfunction, particularly through enhanced monoamine oxidase (MAO) activity and associated increases DA turnover [65]. Upregulated MAO activity is directly linked to central insulin resistance, leading to decreased DA signaling as well as increased brain mitochondrial dysfunction and oxidative stress, specifically in the NAcc and striatum [65]. Therefore, lower basal dopaminergic activity in the mesolimbic pathway of OLETF rats may also be partially due to hyperinsulinemia.

Our finding of hyperleptinemia in OLETF rats is consistent with previous work [66,67]. Interestingly, this strain does not develop central or peripheral leptin resistance, suggesting that increases in serum leptin is likely a compensation to hyperphagia-induced obesity [68]. Recent studies show that the mechanisms contributing to the obesity of OLETF rats are leptin-independent [67,69,70,71] and involve hypothalamic NPY neurons [72]. We found no difference in corticosterone levels between the two strains (Figure 5). Corticosterone increases DA signaling and drug-seeking behaviors, particularly in the NAcc [73]. Previous studies have shown acquired adrenal insufficiency, with inverse correlations between serum corticosterone and serum leptin in OLETF rats as compared to LETO [74]. However, these reported changes in corticosterone levels between strains were age specific, which we did not test in our study.

## 4. Materials and Methods

### 4.1. Animals

Fourteen-week-old male OLETF and LETO rats were obtained as a generous gift of the Tokushima Research Institute, Otsuka Pharmaceutical, Tokushima, Japan. All animals were individually housed in mesh-floored, stainless-steel hanging cages in a temperature-controlled vivarium while maintained on a constant 12:12-h light–dark cycle (lights on at 0600). Rats were handled daily for a minimum of 1 week prior to the onset of experimental procedures. Tap water and pelleted standard laboratory diet (4.0 kcal/g, 6.2% kcal from fat, Envigo) were available ad libitum throughout experiments. All animal usage was in accordance with National Institutes of Health guidelines and were approved by the Institutional Animal Care and Use Committee at the Pennsylvania State University College of Medicine.

### 4.2. Substances Used

D-amphetamine (Sigma-Aldrich, St. Louis, MO, USA) was dissolved in physiological saline and injected subcutaneously (s.c.) at a dose of 0.5 mg/kg before placing the animal in the open-field arena or 15 min prior to the presentation of sucrose in intake tests. Sucrose (Fisher Scientific, Hampton, NH, USA) was diluted in filtered tap water and used at a concentration of 0.3 M (~10% *w/v*) that is palatable to rats.

### 4.3. Experimental Groups and Procedures

Ten OLETF and five LETO rats with an average body weight of 335.6 ± 8.2 g and 293.6 ± 4.9 g, respectively, at the beginning of the experiments were used for activity and sucrose intake testing for 3 weeks prior to the AMPH testing protocol as follows: OLETF rats with ad libitum sucrose access (OA, *n* = 5); OLETF rats with restricted access to sucrose (daily amount of sucrose was clamped to the average intake of LETOs; OR, *n* = 5); and LETO rats with ad libitum sucrose access (LE, *n* = 5). An additional group of age- and weight-matched naïve OLETF (*n* = 10) and LETO (*n* = 10) rats were used for binding and hormonal assays.

### 4.4. Assessment of Locomotor Activity and Sucrose Tests

Open-field tests were conducted using an automated activity monitoring system (TruScan, Coulbourn Instruments from Harvard Bioscience, Inc., Holliston, MA, USA). Testing was conducted before, during, and after the 3-week sucrose access period. Before sucrose access (week 0–1), animals were tested for basal and AMPH-induced motor activity. For this, animals first underwent three 20 min baseline tests for three consecutive days followed by an AMPH test. On days 1, 3, and 5, the animals received s.c. saline injection immediately before their placement in the open-field arena. Day 1 served as baseline, whereas days 3 and 5 served as “wash-out” periods for AMPH injections. On day 2 and 4, the animals were injected with 0.5 mg/kg dose of AMPH. During sucrose access (week 2–4), we tested the effect of 0.3 M sucrose on basal and AMPH-induced activity at 1 and 3 weeks post sucrose access following the same protocol used prior to sucrose access. Sucrose and food intake, as well as body weight, were also recorded during this period. After locomotor activity tests during sucrose exposure, animals were maintained in their individual home cages for one week (week 5–6) with no treatment or testing except daily readings of intakes and body weight. Then, we tested the effect of 1 week of sucrose withdrawal access (week 7) on basal and AMPH-induced activity. Throughout the experiments, all animals were maintained on ad libitum access to chow and water in their home cages, whereas no food and water was available in the open-field arena during testing. Locomotor activity (horizontal beam crossing) was collected in 5 min bins and expressed as percent baseline where locomotor activity associated with each habituation bin was divided by the average habituation activity. Similarly, the locomotor activity associated with the saline injection was normalized to the 20 min of the habituation period when locomotor activity associated with the novel environment had stabilized. Increased activity attributed to AMPH administration is expressed as percent change from the average locomotor activity after the saline injection prior and after AMPH injection.

### 4.5. Dopamine Transporter Binding Assay

After rats were decapitated and blood was collected for hormone analyses, brains were removed and immediately immersed in −40 °C isopentane (2-methylbutane) and stored at −80 °C. The brains were sectioned on a cryostat in the coronal plane at 20 μm and thaw-mounted on poly-lysine coated slides. The brain regions examined were from the dorsal and ventral striatum (1.7–1.1 mm from bregma, inclusive of the NAcc, both the shell and core subregions) and from the mesencephalon [−5.6 to −6.1 mm from bregma, inclusive of the medial ventral tegmental area (VTA) and the substantia nigra (SN)]. Sections from each brain region were mounted on multiple slides, with serial sections distributed across the slides so that each slide had every fourth section. Adjacent slides were selected and binding values from two adjacent sections (representing a distance of 80 μm) were analyzed and averaged for each structure. One slide, each with two sections rostral and two sections caudal to each target region, was stained for cresyl violet and served to determine anterior–posterior coordinates for selection of proper slides and slices for analytical comparisons. For the autoradiography assay, we used our previously published protocol [75]. Slides were thawed for at least 30 min and then incubated for 90 min at 4 °C in a buffer solution that contained a protease inhibitor cocktail that consisted of 25 mg chymostatin, 54 μM leupeptin, 100 μM EGTA, 100 μM EDTA in 0.05 M dibasic/monobasic phosphate buffer with 20 mg BSA (Sigma, St. Louis, MO, USA), and 32.5 pM of the radioligand^125^I-labeled RTI-55 (2200 Ci/mol; DuPont NEN, Boston, MA, USA). One buffer bath contained the radioligand and 1 μM of paroxetine (GlaxoSmithKline, Pittsburgh, PA, USA) to block RTI-55 binding to the serotonin transporters. To determine nonspecific binding, the other buffer bath contained the radioligand and both 1 μM of GBR-12935 (a dopamine-selective uptake inhibitor, 1-[2-(diphenylmethoxy)-ethyl]-4-(3-phenylpropyl) piperazine, Sigma) and 1 μM paroxetine. Immediately after the incubation, the labeled sections were washed three times for 20 min each in 0.05 M ice-cold PBS buffer. Slides were then dipped in double-distilled water and dried overnight at room temperature. Subsequently, the slides were placed in a cassette with ^125^I microscale standards (Amersham, Arlington Heights, IL, USA) and opposed to Kodak biomax MR film (Eastman Kodak, Rochester, NY, USA) for 30 h.

### 4.6. Quantitative Analysis

Immediately after the exposure time elapsed, the films were developed using standard photographic procedure. Film images were captured and digitized with a Howtek scanner (MultiRad 850, Howtek, Hudson, NH, USA). Tissue images were quantitated by a densitometry procedure using microscales for 125I to generate a standard curve. Quantitative analysis was performed with the computer-compatible analytical imaging station (Imaging Research, St. Catharines, ON, Canada) software. Binding was assessed for a target region unilaterally in all tissues. Background and nonspecific or sense binding or both were subtracted from all assayed tissue, as appropriate.

### 4.7. Plasma Hormone Assays

Blood was collected in EDTA vacutainer tubes from overnight-fasted animals. After 20 μL was removed for blood glucose assay (Elite Glucometer, Bayer, Elkhart, IN, USA), the remainder of the blood sample was gently agitated and maintained on ice until centrifugation at 3000 rpm for 10 min. Plasma was then distributed into three microcentrifuge tubes (Fisher Scientific, Pittsburgh, PA, USA) and stored at −80 °C until assayed. Plasma insulin, leptin, and corticosterone were measured using standard enzyme linked immunosorbent assay kits (MyBioSource; San Diego, CA, USA and Enzo Life Sciences; Farmingdale, NY, USA respectively) following manufacturer’s instructions.

### 4.8. Statistical Analyses

One-way ANOVAs were performed for comparisons of baseline parameters between strains. The effect of drug injection, strain, and drug/strain interaction on behavioral parameters in all experiments was analyzed with two-way ANOVAs. Newman–Keuls post hoc tests were performed when the ANOVA showed significant difference. Binding density and blood glucose and hormone levels in OLETF and LETO rats were compared using one-way ANOVA. All data were expressed as means ± SEM. Differences were considered statistically significant if *p* < 0.05. Statistical analyses were computed with Statistica 6.0 software (Tulsa, OK, USA).

## 5. Conclusions

Our study shows that OLETF rats have lower DA transmission and an exaggerated response to natural (sucrose) and pharmacological (AMPH) stimulation than non-mutant controls. These dopaminergic deficits may be responsible for their compensatory regulation, resulting in excess consumption and preference for palatable meals and subsequent body weight gain.

## Figures and Tables

**Figure 1 ijms-24-09773-f001:**
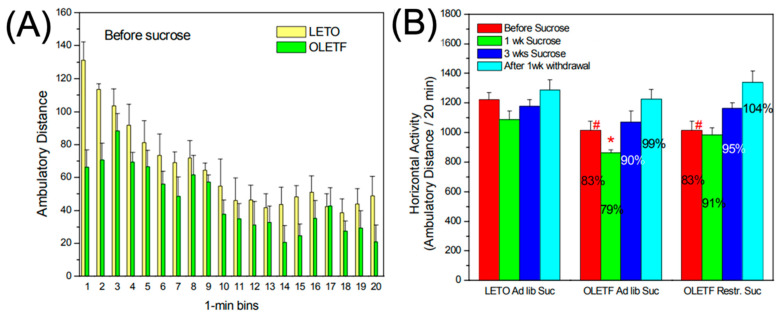
Baseline locomotor activity prior to sucrose administration (**A**). Prior to sucrose administration ambulatory distance was measured in OLETF and LETO rats over 1 min intervals in daily 20 min sessions (pooled data for the 3 sessions are shown). (**B**) Ambulatory distance over a 20 min period in the 3 experimental groups (LETO, OLETF ad lib, and OLETF restricted rats) prior to sucrose administration (red bars), 1 week after sucrose administration (green bars), and 3 weeks after sucrose administration (dark blue bars), as well as 1 week after the commencement of a sucrose withdrawal period. The percentage values overlayed on the bars represent comparisons to the LETO rats’ baseline activity. * *p* < 0.01, # *p* < 0.05.

**Figure 2 ijms-24-09773-f002:**
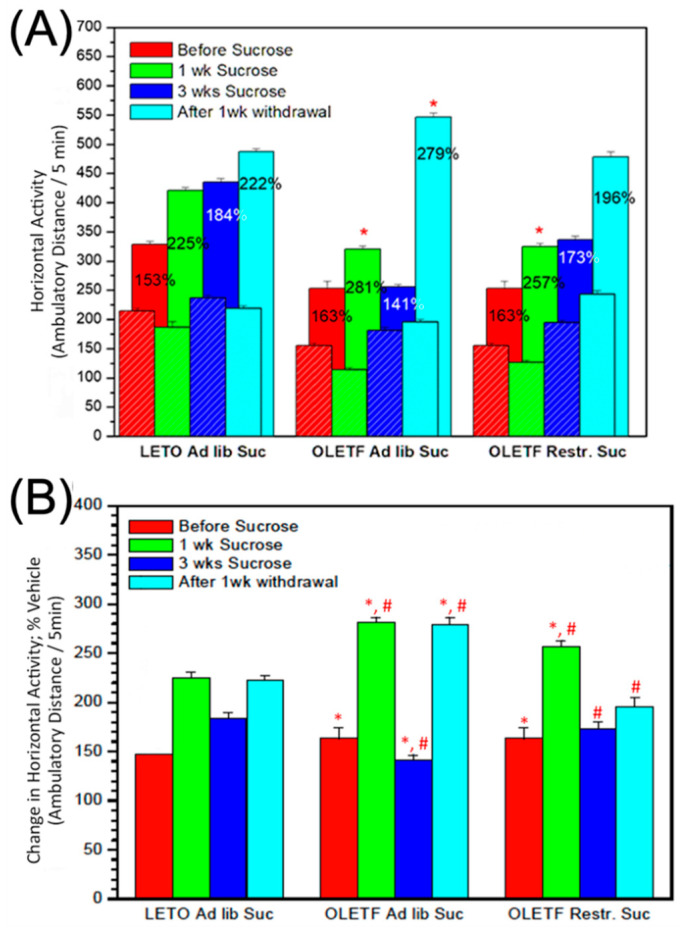
Ambulatory distance in response to AMPH administration (0.5 mg/kg) over a 5 min period prior to sucrose administration, 1 week after sucrose administration, 3 weeks after sucrose administration, and 1 week after the first week withdrawal between LETO, OLETF ad lib, and OLETF restricted rats. (**A**) Horizontal activity in response to AMPH and saline vehicle in the LETO, OLETF ad lib, and OLETF restricted rats. The percentage values overlayed on the bars represent comparisons to saline injections. (**B**) Changes in horizontal activity 15 min after AMPH. * LETO compared to OLETF and OLETF restricted; # OLETF with ad libitum sucrose compared with OLETF with restricted sucrose access.

**Figure 3 ijms-24-09773-f003:**
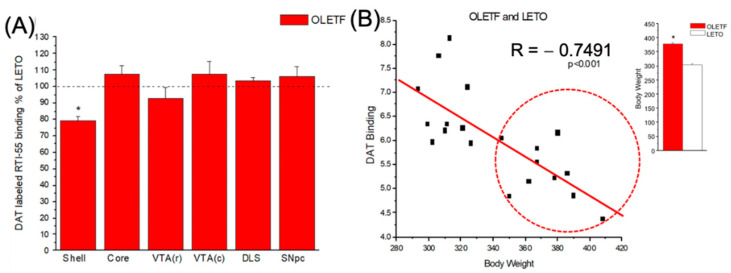
DAT binding assays. (**A**) DAT labeled ^125^I-RTI-55 binding absolute density values (nCi/mg tissue equivalent) for OLETF and LETO, respectively, are as follows: NAcc shell; NAcc core; dorsal lateral striatum (DLS); ventral tegmental area (VTA); pars compacta of substantia nigra (SNpc). (**B**) Correlation between DAT binding and body weight. Correlation between DAT binding and body weight in OLETF (circled with dashed red line) and LETO rats. Regression line is for all data points denoting a significant negative correlation. R = −0.745, *p* < 0.001. * denotes statistical significance.

**Figure 4 ijms-24-09773-f004:**
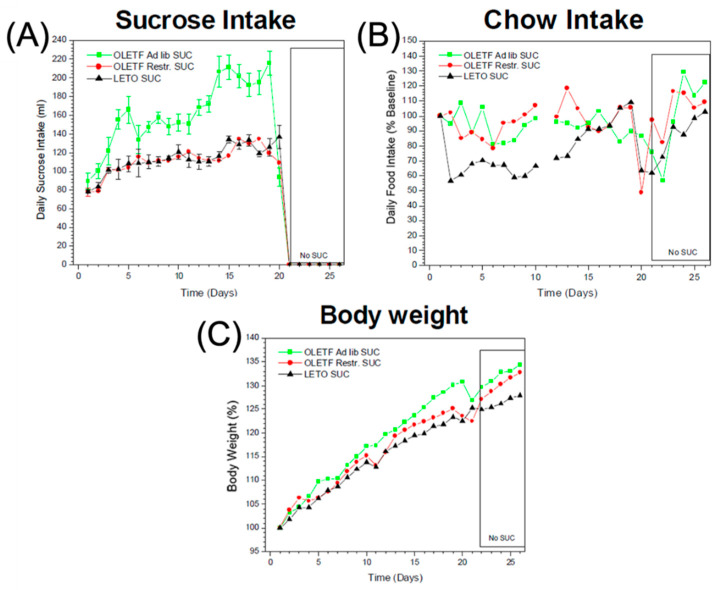
Sucrose and chow intake and changes in body weight over 20 days in OLETF ad libitum fed, OLETF restricted sucrose fed, and LETO sucrose fed rats and 5 days of sucrose withdrawal. (**A**) Sucrose intake (mL/24 h); (**B**) chow intake (g/24 h); (**C**) change in body weight (percent of baseline).

**Figure 5 ijms-24-09773-f005:**
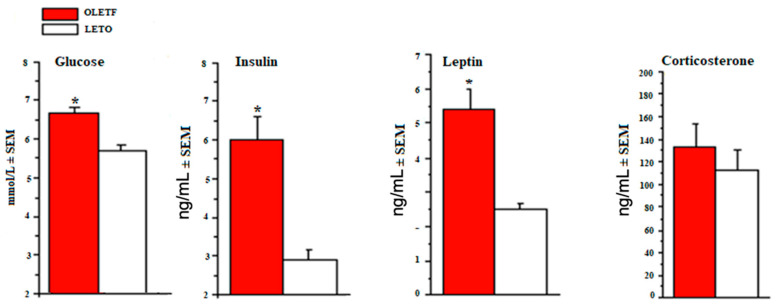
Blood glucose (mmol/L), insulin (ng/mL), leptin (ng/mL), and corticosterone (ng/mL) values in relatively age-matched OLETF and LETO rats. * Statistically different between strains (*p* < 0.05).

## Data Availability

The data presented in this study are available in the article and on request from the corresponding authors.

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
