# Peer review of "Reduced Striatal Dopamine Transporter Availability and Heightened Response to Natural and Pharmacological Stimulation in CCK-1R-Deficient Obese Rats"

_ijms, 2023, doi:10.3390/ijms24119773_

Round 1

Reviewer 1 Report

This work is interesting and seems to be continuation of previous studies. Authors show that OLETF rats lacking cholecystokinin receptor type-1 are hyperphagic, have reduced DA neurotransmission. DA deficit may be responsible for excess consumption and body weight gain. Introduction contains background and aim of the study. Methods are sufficiently described. However, results presentation needs correction. All figures are too small, not clear, lettering too small. I suggest to present the  data which is time dependent not with bars but as linear plot. Importantly, OLETF rats should be presented with matching LETO in panels representing  different  treatments (basal, ad libitum, restricted). In vertical axis mean+/- SEM should be removed and replaced to figure legends. Numbers in the bars are not readable (figure 1A). Dosing in figure 2A, 2B is in figure legend as well as all other information is doubled in figures!

Discussion is too long with to detailed quoting of other authors results.

English needs correction.

Author Response

This work is interesting and seems to be continuation of previous studies. Authors show that OLETF rats lacking cholecystokinin receptor type-1 are hyperphagic, have reduced DA neurotransmission. DA deficit may be responsible for excess consumption and body weight gain. Introduction contains background and aim of the study. Methods are sufficiently described. However, results presentation needs correction. All figures are too small, not clear, lettering too small. I suggest presenting the data which is time dependent not with bars but as linear plot. Importantly, OLETF rats should be presented with matching LETO in panels representing different treatments (basal, ad libitum, restricted). In vertical axis mean+/- SEM should be removed and replaced to figure legends. Numbers in the bars are not readable (figure 1A). Dosing in figure 2A, 2B is in figure legend as well as all other information is doubled in figures!

Discussion is too long with to detailed quoting of other authors results.

Thank you for your constructive comments which helped improve our paper.

We have re-made all figures to improve presentation and clarity. Duplicate text from the graphs was removed and presented only in the Figure legends, as requested. Means +/- SEM from the Y axis label has also been removed. We had considered the reviewer’s suggestion of presenting the data as a function of sucrose access. However, when plotted the data comparing the 3 experimental animal groups across 4 time periods (i.e., prior to sucrose, one week post sucrose, three weeks post sucrose and one week after withdrawal), the graphs are too cluttered and it is more difficult to visually distinguish the main effects. Therefore, for clarity and better visualization of the data without compromising the presentation of the main effects, we elected to retain the figures in the present format (i.e., sucrose access across the three experimental groups). Finally, we have revised and significantly shortened the discussion, as suggested.

Reviewer 2 Report

Hamamah et al., want to examined the severity of the dopaminergic dysfunction in OLETF rats exposed to chronic sweet stimulation, by measuring basal and amphetamine (AMPH)-induced locomotor activity in prediabetic OLETFs compared to age-matched LETOs. They used autoradiography (ARG) to quantify DAT availability in the NAcc and additional nigrostriatal brain areas.

The manuscript contains numerous errors. The discussion is long and shows many other reports, contains a lot of speculation. The data presented in the paper do not show what is discussed in the discussion.

The manuscript needs a lot of improvement.

Introduction is quitable.

Methodology was written rather simple and understandable, but it needs some improvement and clarification.

What does the term "trunk blood" mean? How was blood colected? Were the animals pre-fasted to eliminate postprandial differences in glucose and insulin levels between individuals?

On what basis do the authors call the condition of rats pre-diabetic?

Study groups are small, n=5. Data should be presented as means +/- SD, not SEM, as SD shows the distribution in the population and SEM only in a given experiment.

The results are described in many places incomprehensibly and confusingly. There are no explanations regarding which groups statistical significance is indicated.

The Figures are not very clear and the content is hard to read. Please correct.

Although the description of the results (lines 213-214) says: "Prior to sucrose access basal locomotor activity was mildly reduced in OLETFs (-17%, 213 p < 0.05) compared to LETOs (Fig. 1A)", in Fig. 1A did not marked statistical significance. Please add.

Further, the authors wrote: "Sucrose access exerted a biphasic effect on basal activity in both strains, i.e., reduced activity for ~1wk followed by normal or increased activity throughout weeks 2 and 3." But, in LETO strain the reduction locomotor activity for 1 wk, is not significant - in Fig 1b (part first), no stars.

There is no explanation of which groups are compared in the figures. Between which groups and times are there statistical significances?

Why was the LETO group not compared with OLFTrestr.suc when determining the effects of sucrose and withdrawal on AMPH-induced locomotor activity??

Marking the significance with the letters "a, b" and italic and roman (Fig.2) - are illegible. please change.

Line 274 misspelled Figure 5 instead of Figure 4C.

Despite the limited availability of sucrose in the OLFT rest. group, weight gains were comparable to those in the OLFT group with free access to sucrose... (Fig 4c), so did sucrose intake have no effect on body weight? how to explain it?

What was the total intake of kcal over the period of the experiment - three weeks, in each group. Please provide these results.

Why do the authors write about calorie compensation/non-compensation of rats, giving only the results of the amount of food intake in g, and not the amount of calories consumed (from intake from feed and sucrose)? Please correct, add data, change so that the description agrees with the experience.

Authors wrote: "When sucrose was removed, both OLETF groups lost less weight than LETOs." I don't see any calculation of this relationship/dependence in the paper. Please count and add detailed results.

What do the results presented in section 3.5, showing the levels of glucose, insulin, leptin and corticosterone in two strains of rats that did not participate in the experiment, do not consume sucrose, contribute to this topic?

In lines 420, 436, an incorrect figure number was given.

The results presented in subsection 3.5 and the discussion in subsection 4.5 - do not fit the topic of the work. I don't understand the need to list them and discuss them.

The discussion is complicated. It includes a lot of off-topic discussion. The data obtained in the work do not allow for unambiguous confirmation of the hypotheses. The discussion is based on numerous data from other experiments, without indicating the value of the data obtained in this work.

In the discussion, other, earlier studies are cited, the results of which have already shown the relationships presented in this experiment. So it's hard to see what's new in this experience. The discussion contains a lot of speculation and little experience confirming the quoted theories and dependencies (e.g. about CKK1 and CKK2 transmisions).

Only the results obtained in this work should be discussed in the discussion.

Author Response

Hamamah et al., want to examined the severity of the dopaminergic dysfunction in OLETF rats exposed to chronic sweet stimulation, by measuring basal and amphetamine (AMPH)-induced locomotor activity in prediabetic OLETFs compared to age-matched LETOs. They used autoradiography (ARG) to quantify DAT availability in the NAcc and additional nigrostriatal brain areas.

The manuscript contains numerous errors. The discussion is long and shows many other reports, contains a lot of speculation. The data presented in the paper do not show what is discussed in the discussion.

Thank you for your constructive comments which helped improve our paper.

The manuscript needs a lot of improvement.

Introduction is quittable.

Methodology was written rather simple and understandable, but it needs some improvement and clarification.

#1 What does the term "trunk blood" mean? How was blood collected? Were the animals pre-fasted to eliminate postprandial differences in glucose and insulin levels between individuals?

Response: We have removed the word “trunk”. Animals were decapitated and fasted overnight. This information has been included in the Methods (line 172 and 213).

#2 On what basis do the authors call the condition of rats pre-diabetic?

Response: Previous studies from our lab and others have shown that OLETF rats spontaneously develop diabetes including symptoms of polydipsia, polyuria, diabetic nephropathy, and other complications of diabetes mellitus. Between 12 and 20 weeks, these rats exhibit mild obesity and hyperinsulinemia which soon progresses to hyperglycemia and complications of diabetes mellitus (PMID: 33028888, 19106213). Therefore, these rats have been used as a prediabetic and diabetic animal model in various studies.

#3 Study groups are small, n=5. Data should be presented as means +/- SD, not SEM, as SD shows the distribution in the population and SEM only in a given experiment.

Response: We appreciate the reviewers’ suggestion. However, we elected to present the data means with SEMs which is the standard in these types of studies. The SDs estimate the variability in the study sample, while the SEMs estimates the precision and uncertainty of how the study sample represents the underlying population, which we believe is more appropriate for our study.

#4 The results are described in many places incomprehensibly and confusingly. There are no explanations regarding which groups statistical significance is indicated.

Response:  We have clarified statistical significance, where appropriate, both in the text as well as figures.

#5 The Figures are not very clear, and the content is hard to read. Please correct.

Response: We have re-made all figures to improve clarity and presentation

#6 Although the description of the results (lines 213-214) says: "Prior to sucrose access basal locomotor activity was mildly reduced in OLETFs (-17%, 213 p < 0.05) compared to LETOs (Fig. 1A)", in Fig. 1A did not marked statistical significance. Please add.

Response: Fig. 1A depicts ambulatory distance over 1-min interval in 20-min sessions, therefore adding a statistical significance symbol is impractical. The data presented and the statistical analyses includes pooled data from 3 sessions between the two animal groups.

#7 Further, the authors wrote: "Sucrose access exerted a biphasic effect on basal activity in both strains, i.e., reduced activity for ~1wk followed by normal or increased activity throughout weeks 2 and 3." But, in LETO strain the reduction locomotor activity for 1 wk, is not significant - in Fig 1b (part first), no stars.

Response: We added a line to note that in the LETO group, although showing biphasic effect following sucrose access, this was not statistically significant like in the OLETF ad-libitum access. This was also reflected briefly in the discussion (under section 4.2)

#8 There is no explanation of which groups are compared in the figures. Between which groups and times are there statistical significances?

Response: We have added missing statisctical symbols and, where appropriate, we have clarified it in the Figure legends.

#9 Why was the LETO group not compared with OLFT restr.suc when determining the effects of sucrose and withdrawal on AMPH-induced locomotor activity??

Response: Both OLETF as well as OLETF restr. Sucr have been compared with LETO. We added this information in the Figure legends.

#10 Marking the significance with the letters "a, b" and italic and roman (Fig.2) - are illegible. please change.

Response: We have replaced “a and b” with “*” and “#”

#11 Line 274 misspelled Figure 5 instead of Figure 4C.

Response: Thank you for noticing this. We have added the correct figure number to this line.

#12 Despite the limited availability of sucrose in the OLFT rest. group, weight gains were comparable to those in the OLFT group with free access to sucrose... (Fig 4c), so did sucrose intake have no effect on body weight? how to explain it?

Response: As noted in Fig. 4C, OLETF group with clamped sucrose access to LETO have reduced body weight compared to OLETF with ad libitum sucrose access, however this difference was not statistically different. This can be explained by compensatory response. i.e higher total intake (chow plus sucrose) of OLETF compared to LETO which is well documented in this strain.

#13 What was the total intake of kcal over the period of the experiment - three weeks, in each group. Please provide these results.

Response. For consistency and to eliminate confusion, we have removed the references to calorie consumption, since there was no statistical difference between raw intake and calories between the experimental groups.

#14 Why do the authors write about calorie compensation/non-compensation of rats, giving only the results of the amount of food intake in g, and not the amount of calories consumed (from intake from feed and sucrose)? Please correct, add data, change so that the description agrees with the experience.

Response. We have corrected the discrepancies and only referred to food intake and removed references to caloric intake.

#15 Authors wrote: "When sucrose was removed, both OLETF groups lost less weight than LETOs." I don't see any calculation of this relationship/dependence in the paper. Please count and add detailed results.

Response: This sentence has been removed from the discussion.

#16 What do the results presented in section 3.5, showing the levels of glucose, insulin, leptin and corticosterone in two strains of rats that did not participate in the experiment, do not consume sucrose, contribute to this topic?

Response: Assessing the levels of glucose, insulin, leptin and corticosterone in two strains of rats is relevant as it links the dopaminergic changes as assessed through DAT binding data with behavioral data in response to pharmacological (AMPH) and sucrose stimulation. This has been presented in the Discussion section.

#17 In lines 420, 436, an incorrect figure number was given.

Response: Thank you for noticing this. We have added the correct figure numbers to these lines

#18 The results presented in subsection 3.5 and the discussion in subsection 4.5 - do not fit the topic of the work. I don't understand the need to list them and discuss them.

Response: As noted above the hormonal data and changes in the dopamine transmission are intrinsically linked and well documented in this strain. Therefore, presenting this data helps the readers in understanding the mechanisms responsible for excess consumption and food preference in this strain.

#19 The discussion is complicated. It includes a lot of off-topic discussion. The data obtained in the work do not allow for unambiguous confirmation of the hypotheses. The discussion is based on numerous data from other experiments, without indicating the value of the data obtained in this work.

Response. Thank you for your constructive comments. We have revised and shortened the Discussion section significantly and focused on the evidence pertinent to our main hypotheses tested.

#20 In the discussion, other, earlier studies are cited, the results of which have already shown the relationships presented in this experiment. So it's hard to see what's new in this experience. The discussion contains a lot of speculation and little experience confirming the quoted theories and dependencies (e.g. about CKK1 and CKK2 transmissions).

Response: Given the complexity of the mechanisms leading to hyperphagia, obesity and diabetes in this animal model, it is incumbent that we briefly discuss such mechanisms and place our findings within the context of current literature. In the process, we have removed extraneous information and significantly revised the discussion.

#21 Only the results obtained in this work should be discussed in the discussion.

See response to #20.

Round 2

Reviewer 2 Report

The authors have improved the paper enough that it can now be accepted for publication.